

# Advances in OH reactivity instruments for airborne field measurements

Hendrik Fuchs[1,2], Aaron Stainsby[1], Florian Berg[1], René Dubus[1], Michelle Färber[1],
Andreas Hofzumahaus[1], Frank Holland[1], Kelvin H. Bates[3,4,a], Steven S. Brown[3,5], Matthew M. Coggon[3],
Glenn S. Diskin[6], Georgios I. Gkatzelis[1], Christopher M. Jernigan[3,4], Jeff Peischl[3,4,b], Michael
A. Robinson[3,4], Andrew W. Rollins[3], Nell B. Schafer[3,4], Rebecca H. Schwantes[3], Chelsea E. Stockwell[3],
Patrick R. Veres[3,c], Carsten Warneke[3], Eleanor M. Waxman[3,4], Lu Xu[3,4,d], Kristen Zuraski[3,4],
Andreas Wahner[1], and Anna Novelli[1]

[1]Institute of Climate and Energy Systems, ICE-3: Troposphere, Forschungszentrum Jülich GmbH, Jülich, Germany
[2]Department of Physics, University of Cologne, Cologne, Germany
[3]NOAA Chemical Sciences Laboratory, Boulder, Colorado, USA
[4]Cooperative Institute for Research in Environmental Sciences, University of Colorado Boulder, Boulder, Colorado, USA
[5]Department of Chemistry, University of Colorado, Boulder, Colorado, USA
[6]NASA Langley Research Center, Hampton, Virginia, USA
[a]now at: Department of Mechanical Engineering, University of Colorado, Boulder, Colorado, USA
[b]now at: NOAA Global Monitoring Laboratory, Boulder, Colorado, USA
[c]now at: National Center for Atmospheric Research, Boulder, Colorado, USA
[d]now at: Department of Energy, Environmental and Chemical Engineering, Washington University in St. Louis, Missouri,
USA

**Correspondence:** Hendrik Fuchs (h.fuchs@fz-juelich.de)

**Abstract.** Hydroxyl radical (OH) reactivity, which is the inverse lifetime of the OH radical, provides information on the burden of air pollutants, since almost all air pollutants react with OH. OH reactivity measurements from field experiments can help to identify gaps in the measurement of individual reactants and serve as a proxy for the potential formation of secondary pollutants, including ozone and particles. However, OH reactivity is not regularly measured specifically on airborne

platforms due to the technical complexity of the instruments and/or the need for careful instrumental characterisation to apply accurate correction factors to account for secondary chemistry in the instruments. The method used in this work, based on the time-resolved measurement of OH radicals produced by laser flash photolysis in a flow tube, does not require corrections as secondary chemistry in the instrument is negligible for typical atmospheric conditions. However, the detection of OH radicals by laser-induced fluorescence is challenging. In this work, an OH reactivity instrument has been further developed specifically

for airborne measurements. The laser system used to detect the OH radicals has been simplified compared to previous setups, thereby significantly reducing the need for user interaction. The improved sensitivity allows measurements to be made with high time resolution on the order of seconds with a measurements precision of $0.3\,\mathrm{s}^{-1}$. The OH reactivity measurements were validated by using a propane gas standard, which allowed the determination of the reaction rate constant of the OH reaction with propane. The values are in excellent agreement with literature recommendations within a range of 4 to 8 %. Deviations are

well within the combined uncertainties. The accuracy of the OH reactivity measurements is mainly limited by the determination



of the instrumental zero, which has a typical maximum uncertainty of $0.5\,\mathrm{s}^{-1}$. The high sensitivity of the improved instrument facilitates the data acquisition on board an aircraft as demonstrated by its deployment during the AEROMMA campaign in 2023.

# 1 Introduction

A large number of inorganic and organic species are emitted into the atmosphere from anthropogenic and biogenic sources, making it difficult to detect all of them simultaneously in field experiments and in air quality monitoring stations (Goldstein and Galbally, 2007). All these compounds are chemically transformed in the atmosphere by oxidation reactions and thereby form secondary pollutants such as ozone and particles. Most of them react with the primary oxidant in the atmosphere, the hydroxyl radical (OH), which is formed mainly by the photolysis of ozone and the subsequent reaction of the excited oxygen
atom ($O(^1D)$) with water vapour. Therefore, atmospheric measurements of the OH reactivity, the inverse lifetime of the OH radical, can be used as a proxy for the total amount of chemically active compounds. The OH reactivity ($k(OH)$) is defined as:

$$k(\mathrm{OH}) = \sum_i \left( k_{\mathrm{OH}+\mathrm{X_i}} [\mathrm{X_i}] \right) \tag{1}$$

where $k_{\mathrm{OH}+\mathrm{X_i}}$ is the OH reaction rate coefficient of the compound $\mathrm{X_i}$ at a concentration of $[\mathrm{X_i}]$. As the OH reactant concen-
trations are weighted by the OH reaction rate coefficient, the OH reactivity describes the total chemical turnover of both, the OH radical and the reactive trace gases, and therefore gives the potential for the formation of secondary pollutants (Whalley et al., 2016; Williams et al., 2016).

OH reactivity has been measured in field campaigns for more than 20 years (Kovacs and Brune, 2001; Yang et al., 2016), providing valuable complementary information to individual trace gas measurements. Measurements of single compounds
could explain the measured OH reactivity in some campaigns (e.g. Mao et al., 2010; Fuchs et al., 2017b) but also large gaps of up to a factor of 2 to 3 between measured OH reactivity and calculations from OH reactant concentrations have been observed, especially in forests (Kovacs et al., 2003; Nölscher et al., 2012). If OH concentrations are measured simultaneously, the total loss rate of OH radicals can be calculated and compared with the production rate of OH radicals. Gaps in the chemical budget of OH radicals have also been observed in several field campaigns (Hofzumahaus et al., 2009; Whalley et al., 2011; Tan et al.,
2018), which were conducted in environments with high loads of organic compounds. However, gaps have also been found in relatively clean rural air (Elshorbany et al., 2012; Cho et al., 2023). In all these field campaigns, the unbalanced OH budget indicates unidentified OH radical sources and an incomplete understanding of the atmospheric radical chemistry (Rohrer et al., 2014). Few OH reactivity measurements have been performed on airborne platforms such as on an aircraft (Mao et al., 2009; Thames et al., 2020) or on a Zeppelin (Kaiser et al., 2015). Instruments measuring OH reactivity have also been used for
laboratory studies of reaction kinetics by measuring gas mixtures containing known concentrations of individual reactants (Sadanaga et al., 2006; Stone et al., 2016; Medeiros et al., 2018; Wei et al., 2020; Berg et al., 2024).





Two main methods have been developed to measure OH reactivity. The comparative reactivity method (CRM) compares the consumption of OH radicals is compared when either an artificially introduced OH reactant that is not typically present in the atmosphere (most commonly pyrrole) or OH reactants in the sampled air react with artificially produced OH radicals in a reaction volume (Sinha et al., 2008). The higher the concentration of OH reactants in the sampled air, the less of the artificial OH reactant is consumed. The artificial OH reactant is most commonly measured by proton-transfer-reaction mass spectrometry (PTR-MS), but gas-chromatography has also been used (Nölscher et al., 2012; Praplan et al., 2017). A challenge of the CRM technique is the need for large corrections to account for secondary chemistry in the reaction volume (e.g. Michoud et al., 2015).

In the flash photolysis and laser-induced fluorescence method, the loss of OH radicals is directly measured in a flow tube through which air containing the OH reactants is sampled (Sadanaga et al., 2004). Some instruments use a movable injector to inject artificially produced OH radicals, allowing the reaction time to be varied (Kovacs and Brune, 2001; Hansen et al., 2014). Most instruments, however, produce OH by flash photolysis of ozone using a short laser pulse at a wavelength of 266 nm from a quadrupled Nd:YAG laser (Sadanaga et al., 2004; Lou et al., 2010). The following OH decay is observed with a high time resolution by laser-induced fluorescence after excitation by a pulsed, high-frequency dye laser system providing radiation at a wavelength of 308 nm. Chemical ionisation mass spectrometry has also been used to detect the OH radicals (Muller et al., 2018).

Instruments for OH reactivity measurements used in field campaigns were compared in chamber experiments in the large outdoor chamber SAPHIR in 2015 and 2016, which allowed for a systematic investigation of the performances of the instruments under controlled conditions (Fuchs et al., 2017a). The results showed that all instruments gave accurate results. Instruments using flash-photolysis and OH detection by laser-induced fluorescence showed the highest precision and accuracy as a high repetition rate of measurements is possible and no corrections are required due to secondary chemistry in the instruments for typical atmospheric conditions. However, as there is no commercial instrument available to detect OH radicals, this method is currently only used by groups that also have instruments to measure atmospheric OH concentrations. The complexity of this method has certainly prevented wider use of this type of instrument.

In this work, it is shown that it is possible to reduce the technical complexity for the flash-photolysis and laser-induced fluorescence method such that it can run autonomously. As a proof-of-concept, the improved instrument was deployed on the NASA DC-8 aircraft during the AEROMMA (Atmospheric Emissions and Reactions Observed from Megacities to Marine Areas) campaign led by the National Oceanic and Atmospheric Administration (NOAA) in summer 2023.

## 2 Measurement of OH reactivity by flash-photolysis and laser-induced fluorescence

The instrument described in this work is based on an instrument that was first used in a field campaign in the Pearl-River-Delta, China (Lou et al., 2010) and in experiments in the SAPHIR atmospheric simulation chamber at Forschungszentrum Jülich, Germany (Fuchs et al., 2013). Previous versions shared several parts, such as the laser system for the OH detection, with an



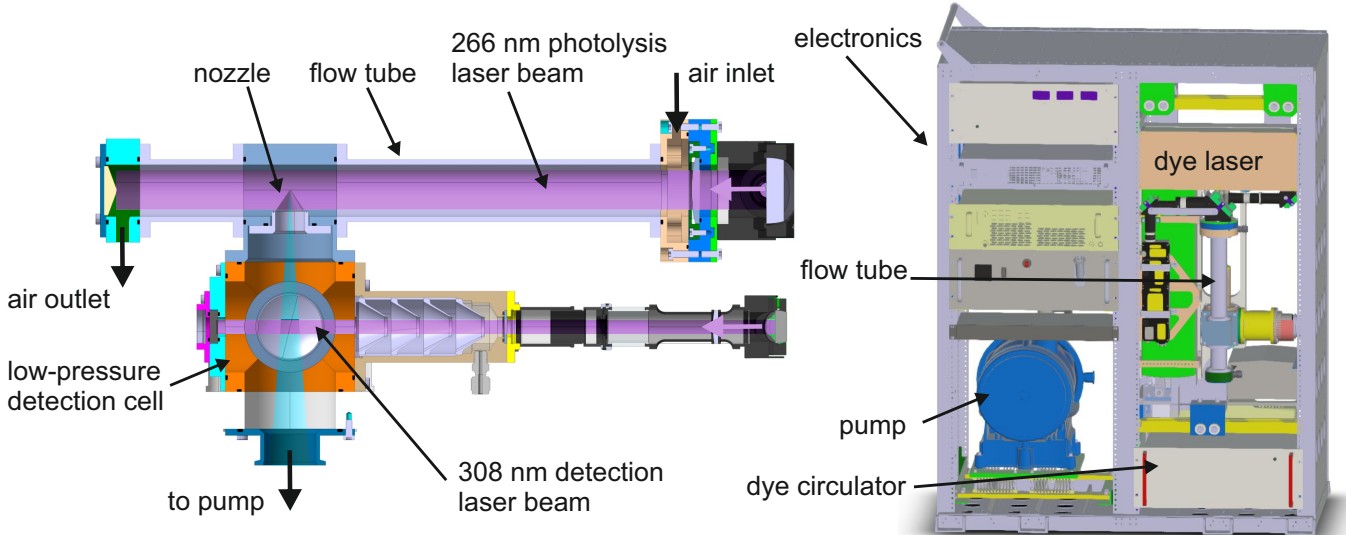

**Figure 1.** Schematic of the flow tube, through which air containing OH reactants flows, and the detection cell for measuring OH radicals by fluorescence. The photolysis and the detection laser beams are directed by turning mirrors into the flow tube and the fluorescence cell, respectively. The laser beams are expanded by lenses to diameters of 30 mm (photolysis laser) and 8 mm (detection laser). All parts of the instrument including all utilities parts and pumps were built in a 19" double-rack for the flights on the NASA DC-8 aircraft during the AEROMMA campaign 2023.

instrument measuring contemporary radical concentrations (Lou et al., 2010). However, the new instrument used on the aircraft

80 was designed to be used as a stand-alone instrument.

The instrument consists of two main parts: (1) a flow tube, through which air containing OH reactants flows continuously and in which a high concentration of OH radicals is generated by ozone photolysis applying a short laser-pulse at a wavelength of 266 nm at a low repetition rate (around 1 Hz), and (2) the detection of OH radicals by laser-induced fluorescence using a laser at a wavelength of 308 nm operated at a high repetition rate (13 kHz) (Fig. 1).

85 Air is sampled in a flow tube through a stainless-steel inlet tube (inner diameter: 8 mm, variable length of up to several metres) coated with SilcoNert ® to minimise losses of reactive species in the inlet. The flow tube made of anodised aluminium has an inner diameter of 40 mm and a length of 50 cm. The flow is controlled by a calibrated mass flow controller (Bronkhorst, Low $\Delta$P Series) downstream of the flow tube that is backed up by a scroll pump (Agilent, IDP-3). The flow rate is chosen such that the residence time of air in the flow tube is approximately two seconds. For ambient conditions, the typical flow rate is

90 between 13 and 20 l/min.

OH radicals are produced in the flow tube by the photolysis of ozone at 266 nm forming excited oxygen atoms O($^1$D), which subsequently react with water vapour forming 2 OH radicals on a time scale of nanoseconds for conditions in the flow





tube:

$$O_3 + h\nu\,(266\,\text{nm}) \quad \rightarrow \quad O_2 + O(^1D) \tag{R1}$$

$$O(^1D) + H_2O \quad \rightarrow \quad 2\,OH \tag{R2}$$

In field experiments, ozone and water vapour concentrations in the sampled ambient air are usually high enough to produce a sufficiently large OH concentration but in laboratory experiments, in which synthetic air is used, ozone and humidity have to be added. For this purpose, oxygen is photolysed by 185 nm radiation from a low-pressure mercury lamp in a custom-build ozoniser. Water vapour is added using either a water bubbler or a controlled evaporator mixing system (Bronkhorst, CEM), in which Milli-Q ® water is evaporated. The CEM system allows precise control of the water vapour mixing ratio. The addition of ozone and water vapour requires the availability of bottled synthetic air. If water vapour and / or ozone are added to the sampled ambient air, the dilution of the ambient air needs to be considered in the evaluation. Sensors measure the pressure (Honeywell, PPT) and relative humidity together with temperature (Vaisala, Humicap) at the outlet of the flow tube.

The 266 nm radiation is generated by a compact quadrupled Nd:YAG laser (Lumibird, Ultra 100) that delivers short laser pulses (10 ns) with a pulse energy of 20 mJ. The laser operates at a low repetition rate (0.93 to 1 Hz). The exact frequency is set to minimise the overhead time between two consecutive OH decay measurements, taking into account the duration over which the OH decay is observed and the time required to transfer the data from the photon counting electronics to the computer. The laser beam is expanded by a lens telescope to a diameter of 30 mm. For typical atmospheric ozone (20 to 50 ppbv) and water vapour (0.2 to 1.8 %) mixing ratios, the initial OH concentration is of the order of a few $10^9\,\text{cm}^{-3}$. The photolysis laser is aligned to illuminate almost the full cross section of the flow tube, so that the OH radicals are approximately homogeneously distributed in the flow tube.

Near the end of the flow tube, a small fraction of the air (1 slm, slm: litres per minute at standard conditions) is sampled into a low-pressure detection cell through a conical nozzle (pinhole diameter: 0.4 mm) that sticks into the centre of the flow tube (Fig. 1). The pressure of the detection cell is typically 2.5 to 4 hPa for atmospheric pressure in the flow tube. In the detection cell, OH radicals are excited at their rotational absorption line $Q_1(3)$ of the $OH(A^2\Sigma, \nu'=0 \leftarrow X^2\Pi, \nu''=0)$ band transition by a short laser pulse (20 ns) at a wavelength of 308 nm. The 308 nm laser radiation is generated by a custom-built dye laser (dye: Rhodamine 519) (Strotkamp et al., 2013) that is pumped by a commercial frequency-doubled Nd:YAG laser (Spectraphysics, Talon). The 616 nm light produced by the dye laser is frequency doubled to the UV by a barium borate (BBO) crystal inside the laser cavity. The average laser power at a frequency of 13 kHz is up to 200 mW. Mirrors guide the laser light to the detection cell. The beam size is expanded to a diameter of approximately 8 mm before entering the detection cell.

Perpendicular to the axis of the air flow and the axis of the laser beam, fluorescence photons are collected by a set of condenser lenses which direct the photons to a microchannel plate photomultiplier (Photek, MCP 325). Opposite of the lens system, a spherical mirror reflects the photons towards the detection system, almost doubling the solid angle from which the photons are detected. As the fluorescence wavelength is resonant to the laser excitation wavelength, the detector is electronically gated while the laser pulse is applied and photon counting starts shortly after with a delay of approximately 100 ns for 500 ns. Single photons are counted by photon counting electronics (Becker and Hickl, PMS 400).



After application of the photolysis laser pulse, the OH (initial concentration $[OH]_0$) reacts away in the flow tube. As the OH reactants have much higher concentrations than the OH radicals, the time behaviour of the OH concentration can be described by a pseudo-first order loss process:

$$[OH](t) = [OH]_0 \exp(-k(OH) t) \tag{2}$$

The corresponding measured photon counts ($N(t)$) includes a constant background signal ($B$), which is mainly caused by scattered laser light and detector noise:

$$N(t) = N_0 \exp(-k(OH) t) + B \tag{3}$$

After the application of the photolysis laser, the decay of the photon counts is recorded for $1\,\mathrm{s}$ with a time resolution of $1\,\mathrm{ms}$.
The OH reactivity is calculated from a single-exponential fit to the measured photon counts using a Levenberg-Marquardt minimisation procedure. Depending on the detection sensitivity of the OH measurement and the OH concentration produced, several measurements are summed or averaged before the fit is applied. A minimum amplitude of the OH decay curve around 40 counts is sufficient to obtain a reliable fit result.

OH radicals are also lost in wall reactions on the surface of the flow tube. This is mainly diffusion limited and the cor-
responding OH decay can be described by a single-exponential function. The resulting instrumental zero-decay rate ($k_0$) is typically between 1 and $3\,\mathrm{s}^{-1}$. Its value needs to be regularly determined by measuring the OH loss rate in pure synthetic air containing only ozone and water vapour. The reactivity from the ozone added to the sampled air is negligible ($< 0.1\,\mathrm{s}^{-1}$).

As discussed in Lou et al. (2010), photolysis of OH reactants do not significantly affect the measurements for typical atmospheric conditions. Deviations from a single-exponential decay of the OH concentration are possible, if OH is produced
from secondary chemistry in the flow tube on the time scale of the OH loss rate (Fuchs et al., 2017a). For example, OH radicals are produced by the reaction of hydroperoxy radicals ($HO_2$) with nitric oxide (NO). However, this can only become relevant under exceptional conditions with high NO concentrations (e.g. $> 20\,\mathrm{ppbv}$, Fuchs et al. (2017a)) and rapid production of $HO_2$ for example in the reaction of OH with carbon monoxide (CO). In most cases, however, regeneration of OH does not play a role even at high NO concentrations, because the overall OH reactivity is typically high at these conditions, so that the OH radical
lifetime is much shorter than the time scale of the OH production, and the OH regeneration is too slow to affect the results. OH can also be regenerated if the OH reaction with the reactant forms an adduct which then can decompose and eliminate an OH radical. An example is the reaction of OH with isoprene hydroxy hydroperoxide (ISOPOOH), which is photochemically produced in the oxidation of isoprene mainly emitted by vegetation (St. Clair et al., 2015). The contribution of such species to the total atmospheric OH reactivity is typically small, so that the underestimation of the OH reactivity due to OH regeneration
in the flow tube is usually negligible.



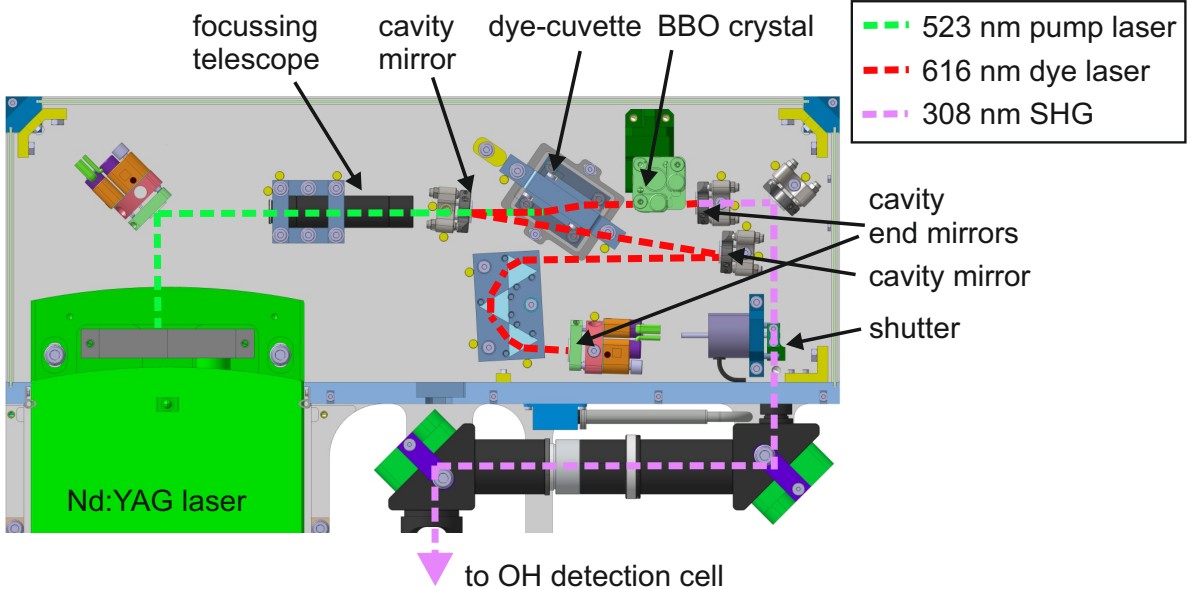

**Figure 2.** Schematic of the simplified dye laser system (wavelength 616 nm) pumped by a commercial Nd:YAG laser (wavelength 532 nm) used for the OH detection. The dye laser cavity consists of the 2 end mirrors and 2 additional mirrors that fold the beam path of the cavity. The dye cuvette is positioned at the Brewster angle. Prisms are used to select the wavelength of the dye laser, which can be tuned by the horizontal position of the end mirror, which is mounted in a motorised mirror mount. The dye laser radiation is frequency converted to a wavelength of 308 nm by a BBO crystal inside the laser cavity. The UV light is directed to the fluorescence cell in a lens tube system via deflection mirrors (Fig. 1).

## 3 Improvements and characterisation of OH reactivity measurements for (airborne) field campaigns

### 3.1 Improvements of the laser stability and sensitivity

In order to measure OH reactivity specifically on an aircraft, the instrument needs to be robust against vibration, pressure and temperature changes. In addition, an autonomous operation is advantageous and a high time resolution in the range of a few seconds is desired, as rapid changes in the ambient reactivity are expected due to the high speed of the aircraft, e.g., when flying through a pollution plume. An instrument meeting these requirements will also be suitable for easy deployment in ground-based field campaigns and may also be used in air quality monitoring stations.

The major challenge of OH reactivity measurements using the direct detection of OH radicals by fluorescence is the detection laser, which is typically a dye laser system (Section 2). In previous versions, the OH reactivity instrument was part of a measurement system that also included OH concentration measurements and therefore shared several parts, e.g. the laser system (Lou et al., 2010). The aim of developing an OH reactivity instrument for aircraft applications was to have a stand-alone instrument. Therefore, the previously developed laser system (Strotkamp et al., 2013) was further developed and optimised for



the measurement of OH reactivity only. A reduced complexity of the system is possible because no absolute OH concentration needs to be measured and the initial OH concentration is high.

For the relative time-resolved OH measurements required in the OH reactivity instrument, the laser wavelength does not need to be tuned on and off the OH absorption line, since background signals, e.g. from detector noise or the fluorescence of species other than OH do not need to be subtracted from the total signal. They only appear as an offset in the OH decay curve (Eq. 2) as long as they do not change over the time of the OH decay and/or are small compared to the OH fluorescence counts. Due to the high OH concentration, also a lower OH detection sensitivity due to a lower OH excitation efficiency than for OH

concentration measurements is acceptable.

In the instrument used for OH concentration measurements, the tuning of the laser wavelength and the narrow spectral width of the laser in the order of the Doppler-broadened OH absorption (approximately 3 GHz) is achieved by a movable etalon in the dye laser resonator (Strotkamp et al., 2013). The alignment is sensitive to temperature variations and vibrations, which can especially occur during flights. As the laser wavelength tuning is not required for OH reactivity measurements, the etalon is

removed from the optical design in the stand-alone instrument. In this laser design, the laser wavelength is determined by the optical path through the prisms in the dye laser cavity (Fig. 2). This results in a broad spectral width of approximately 0.03 nm, much wider than the OH absorption line, so that the OH radical is effectively excited even if the central wavelength drifts slightly, making the setup robust to small changes in the laser alignment.

The peak wavelength of the dye laser can be tuned by the horizontal position of the cavity end mirror of the dye laser, which

is mounted in a motorised mirror mount (Newport, Picomount). Changes are expected for example in the aircraft, as the cabin pressure is reduced after take-off, changing the refractive index of the air and therefore the centre wavelength of the dye laser output. The laser wavelength is monitored by a high resolution spectrometer (Ocean Insight, HR4000, resolution: 0.03 nm), which allows automatic tuning to the OH absorption line by software.

In addition, autonomous and stable operation of the dye laser is achieved by:

– Heating the plate, on which the dye laser cavity is mounted to a slightly higher temperature than the ambient to avoid temperature drifts in the alignment,

– Mounting the mirror that directs the pump laser beam and the BBO crystal in motorised mirror mounts (Newport, Picomount) to remotely and automatically tune and optimise their position for maximum laser output,

– Mounting all other mirrors of the laser cavity in ultra-stable mirror mounts (Thorlabs, Polaris).

These improvements make the OH detection robust to small changes in the cavity alignment, allowing the laser to operate at a high performance without operator intervention. For example, the dye laser power achieved during the deployment on the NASA DC-8 aircraft was at least 150 mW.

The total number of fluorescence photons in the new OH reactivity instrument is maximised by a high laser repetition rate of 13 kHz, 50 % higher than in previous versions of the instrument. This is possible because the pump laser used in this system

(Newport, Talon) delivers a nearly constant pulse energy up to this repetition rate, so that the dye laser power scales with the





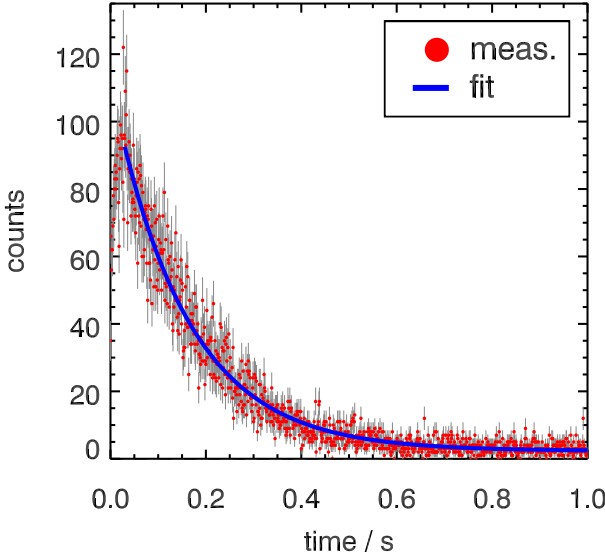

**Figure 3.** Example of a measured OH decay during the AEROMMA campaign on 02 August 2023 at an altitude around 500 m on the NASA DC-8 aircraft. 9 traces were summed before the fit was applied, resulting in a time resolution of 10 s due to some overhead time. A fit of the fluorescence counts to a single-exponential function gives $N(t) = (109.3 \pm 0.2) \cdot \exp(-(6.39 \pm 0.02)s^{-1}t) + (2.33 \pm 0.05)$. The laser power of the 308 nm detection laser was 147 mW.

repetition rate. In addition, potential interferences from artificial OH production in the fluorescence cell, which increase with the laser repetition rate such as photolysis processes of e.g. ozone are not important (Fuchs et al., 2016). They only slightly increase the background signal in the measured OH decay, since it can be assumed that the concentration of a species causing the interference does not change over the time of a OH decay curve of 1 s.

The OH fluorescence yield is further enhanced by back-reflecting a large fraction of the 308 nm laser at the exit of the fluorescence cell using a mirror with a low transmission of 10 %, so that the photon density in the cell is almost doubled. The transmitted laser light is used to monitor the laser power. Again, this is only possible because small artificial photolytic sources of OH inside the fluorescence cell and an increased background signal from laser scattering do not affect the measured OH decay. An example for a OH decay curve from measurements on the NASA DC-8 aircraft is shown in Fig. 3, demonstrating

the high precision of measurements at a high time resolution (here: average of 10 decay curves resulting in a time resolution of 10 s) that could be achieved with the optimised instrument design.

### 3.2    Precision of OH reactivity measurements using the improved design

Figure 4 shows an Allan deviation plot derived from OH reactivity measurements in humidified synthetic air (1.5 % water vapour mixing ratio) with added ozone resulting in a mixing ratio of 60 ppbv ozone. The initial OH concentration is approx-

imately $8 \cdot 10^9 \, \text{cm}^{-3}$ in the flow tube and corresponds to an amplitude of the OH fluorescence signal of 24 counts for one photolysis laser shot. This can be converted to a sensitivity of the OH detection of 0.002 cnts per $10^6 \text{cm}^{-3}$ OH radicals per





mW laser power of the 308 nm detection laser. This number is approximately a factor of 10 lower than the sensitivities achieved in instruments for the measurement of OH radical concentrations (Fuchs et al., 2012) due to the much broader spectral width of the laser used in the new OH reactivity instrument.

The Allan deviation demonstrates a high precision of approximately $0.3\,\mathrm{s}^{-1}$ of the OH reactivity measurement at a time resolution of $1\,\mathrm{s}$. An even higher precision of, e.g. $0.07\,\mathrm{s}^{-1}$ is obtained for an integration time of $10\,\mathrm{s}$. The distribution of OH reactivity measurements (Fig. 4) shows deviations from a Gaussian distribution when individual OH decay curves ($1\,\mathrm{s}$ integration time) are evaluated as seen by the fraction of values that deviate from zero by more than $0.5\,\mathrm{s}^{-1}$. The number of outliers can be significantly reduced if at least 3 OH decay curves are summed before applying the exponential fit because small systematic deviations from a single-exponential behaviour are smoothed out. This demonstrates that an integration time of at least $3\,\mathrm{s}$ is recommended to ensure a statistical error of the OH reactivity measurements.

The produced initial OH concentration in ambient air may be lower than in the laboratory measurements, as the mixing ratios of ozone and water vapour are highly variable. However, the Allan deviation of measurements in synthetic air shows that a high time resolution in the range of seconds can still be easily achieved by summing several traces. The detection limit is much lower than the OH reactivity in ambient air, which typically has minimum values around $1\,\mathrm{s}^{-1}$ even in clean environments (Section 4).

For the evaluation of the data collected during the AEROMMA campaign on board the NASA DC-8 aircraft, typically 5 to 10 OH decay curves were summed before calculating the OH reactivity from the OH decay curve, in order to achieve a sufficiently high precision of the data (e.g. minimum amplitude of 40 counts). As the repetition rate of the photolysis laser beam was 0.93 Hz (Section 2), this results in a time resolution of measurements of 5.5 to 11 s. For conditions of very low water vapour mixing ratios of less than 0.1 % encountered at high altitudes or in dry areas, up to 30 to 40 OH decay curves had to be added up. At even lower water vapour mixing ratios of less than 0.05 % experienced in the free troposphere at altitudes above 8 km, the instrument cannot operate without the addition of water vapour, which was not foreseen in this campaign.

### 3.3 Considerations for the use on an aircraft

When the instrument is operated on board an aircraft, the air is drawn into the flow tube from outside the aircraft, so that the pressure in the flow tube is similar to the ambient pressure, which decreases with altitude. On the NASA DC-8 aircraft, a standard inlet system provided by NASA was used. A restrictor that was part of the inlet system resulted in an approximately 100 hPa higher pressure than ambient pressure in the flow tube during the flight. A reduced air pressure in the flow tube compared to ground conditions has several consequences, some of which require adjustments to the operation of the instrument during the flight:

– In order to keep a similar residence time of the air in the flow tube, the sampling flow rate controlled by a mass flow controller is automatically adjusted, so that the volumetric flow rate becomes similar with the changing pressure. If this was not done, the residence time of the air in the flow tube can become shorter than the time between two successive photolysis laser shots, so that the decay curve would be affected by the flushing out of the OH radicals.




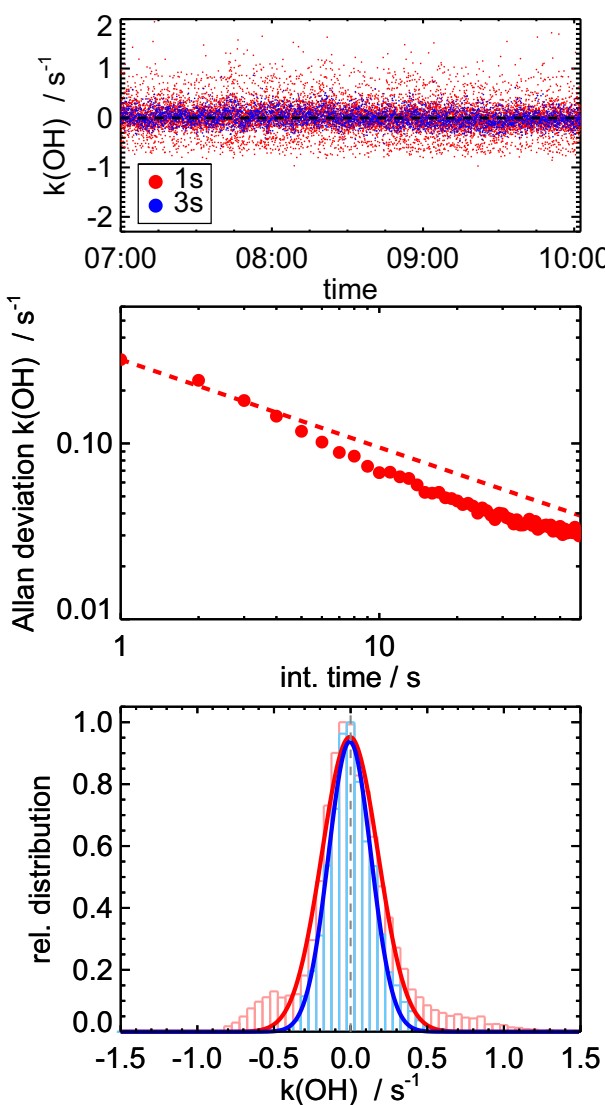

**Figure 4.** Allan deviation plot (middle panel) of OH reactivity measurements (top panel) from measurements after subtracting the zero reactivity value in a mixture of humidified synthetic air (water vapour mixing ratio: 1.5 %) with ozone (60 ppbv). The dashed line gives the Allan deviation expected from Gaussian noise. The distribution (lower panel) of zero measurements for 1 s data (red) shows deviations from a Gaussian distribution (lines) that are reduced when 3 traces are summed before the single-exponential fit is applied (blue).





– The diffusion rate of OH radicals increases, which may affect the zero-decay rate.

– The initial OH concentration in the flow tube is reduced because the concentrations of ozone and humidity is reduced due to the reduced number density of molecules, leading to a lower amplitude of the fluorescence counts per shot of the photolysis laser.

– A critical nozzle is used to sample air from the flow tube into the OH detection cell, which inherently results in a constant
volumetric flow rate into the cell. Since the volume flow rate through the flow tube is also constant, the fraction of air sampled into the detection cell remains independent of the flight altitude.

– As the mass flow sampled into the detection cell decreases, the pressure in the detection cell decreases if the power of the pump downstream of the cell is kept constant, leading to a decrease of the detection sensitivity, if this is optimized for ambient pressure on the ground (typical detection cell pressure: 3 to 4 hPa). The pressure could be increased by
adjusting the flow restriction using a butterfly valve between the detection cell and the pump (Fig. 1). However, only a manual valve was installed in the AEROMMA campaign.

Overall, there is a significant reduction in the detection sensitivity at high altitude of the aircraft, which can be compensated for by a longer integration time. This can typically be accepted as the variability of the OH reactivity is expected to be small at high altitude.

### 3.4    Characterisation of the instrument zero-decay rate

The instrument's zero is the loss of OH radicals in the absence of an OH reactant due to the wall loss of OH (Section 2). It can be described as a pseudo-first order loss process and must be characterised thoroughly for an accurate determination of OH reactivity measurements.

In the laboratory characterisation experiments high purity synthetic air mixed from evaporated liquid nitrogen and oxygen
(purity $> 99.9999\%$, Linde) and ultra-pure Milli-Q $^{®}$ water was used. This makes it unlikely that the zero-decay is caused by the introduction of OH reactants in the laboratory measurements. Tests in the field during the AEROMMA campaign were performed with ultra-pure bottled synthetic air (Linde) and Milli-Q $^{®}$ water, but resulted partly in up to $1\,\mathrm{s}^{-1}$ higher zero values than measured in the laboratory. The exact value depended on the specific synthetic air bottle, so that the higher values were likely due to impurities. Therefore, zero measurements from the laboratory were used for the evaluation of measurements of
the AEROMMA campaign.

The dependence of the zero-decay rate on pressure and humidity was characterised in laboratory measurements. The pressure in the flow tube was varied by inserting a valve in the inlet line acting as a variable flow restrictor. The volume flow rate in the flow tube was kept constant during these tests by automatically adjusting the setpoint of the mass flow controller downstream of the flow tube, as done when operating the instrument on an aircraft. The humidity of the synthetic air was varied by changing
the amount of water that was evaporated in the humidification system (Bronkhorst, CEM).





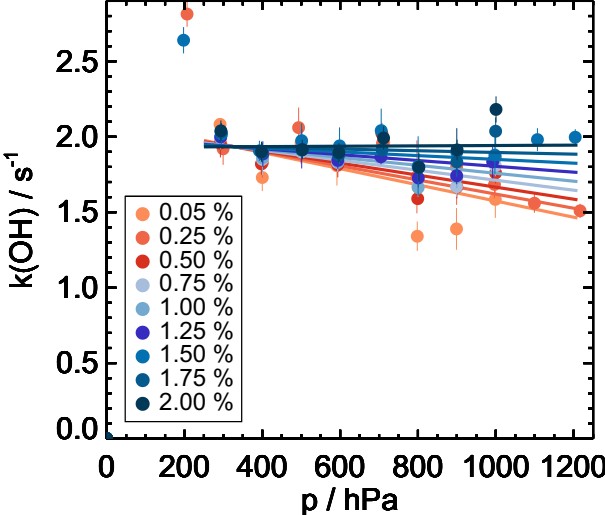

**Figure 5.** Zero-decay values depending on the pressure. The colours give the water vapour mixing ratio during the zero measurement. The dependence can be described by Eq. 4 for pressure values higher than $300\,\mathrm{hPa}$. The measurements of the higher zero decay values below $300\,\mathrm{hPa}$ were reproducible and indicate that other effects like the increased diffusion of OH radicals led to a higher wall loss. During the AEROMMA campaign on the NASA DC-8 aircraft, the pressure in the flow tube was always higher than $350\,\mathrm{hPa}$.

The overall dependence of the zero-decay rate on humidity and pressure is small for pressures higher than $300\,\mathrm{hPa}$ and water mixing ratios higher than $0.5\,\%$ (Fig. 5). The values are between $1.6$ and $1.9\,\mathrm{s}^{-1}$. Only at very low pressures of $200\,\mathrm{hPa}$, lower than the pressure in the flow tube experienced during the AEROMMA campaign ($> 350\,\mathrm{hPa}$), does the zero-decay time increase to values higher than $2.5\,\mathrm{s}^{-1}$. The small dependence of the zero-decay time on pressure (pressure $> 300\,\mathrm{hPa}$) and
water mixing ratio can be expressed as:

$$k_0 = (a_0 + a_1 \cdot p)\,[\mathrm{H_2O}] + a_2 \cdot p + k_0 \tag{4}$$

Fitting the values obtained in the laboratory experiments gives values of the parameters of $a_0 = -0.09\,\mathrm{s}^{-1}\%^{-1}$, $a_1 = 2.7 \cdot 10^{-4}\,\mathrm{s}^{-1}\mathrm{hPa}^{-1}\%^{-1}$, $a_2 = -5.4 \cdot 10^{-4}\,\mathrm{s}^{-1}\mathrm{hPa}^{-1}$, $k_0 = 2.1\,\mathrm{s}^{-1}$. The differences between the parametrisation and the measured values are less than $0.3\,\mathrm{s}^{-1}$, which is within the reproducibility of the zero-decay measurement, giving a lower limit of
the accuracy of the OH reactivity measurements.

The zero-decay rate is most likely caused by the loss of OH radicals on the wall of the flow tube. Due to its high reactivity, it can be assumed that the probability of wall loss on metal surfaces, such as in the OH reactivity instrument, is very high for OH radicals, so that the total loss rate is mainly limited by diffusion (Lou et al., 2010). However, this description only holds if the initial OH concentration is homogeneously distributed in the flow tube. Deviations from this can occur if either the photon
density of the $266\,\mathrm{nm}$ photolysis laser is not homogeneous, or the laser beam is not well aligned.

With decreasing pressure, the diffusion of OH radicals towards the wall of the flow tube increases as the diffusion coefficient is inversely proportional to the pressure. Therefore, an increase in the zero-decay rate is only clearly visible at lower pressures



for low water vapour mixing ratios. The increase in the zero-decay rate with increasing water vapour mixing ratio could be caused by a higher probability of the OH loss at the wall. However, it can also not be fully excluded that there was a small

contamination of the water.

Only one other OH reactivity instrument has been deployed on an aircraft (Mao et al., 2009; Thames et al., 2020). This instrument uses a movable injector inside a flow tube, through which a small amount of humidified air containing OH and $HO_2$ radicals produced by water photolysis at a wavelength of $185\,\mathrm{nm}$ from a mercury lamp is injected. In this instrument, the OH wall loss is assumed to be independent of pressure, but an OH loss due to impurities in the injected air was observed (Mao

et al., 2009). Since the mass flow rate of the injected air was the same at all altitudes but the volume flow rate of sampled air decreased with height, a pressure dependence of the zero-decay rate appeared due to the change in the dilution of contaminants (Thames et al., 2020). The effect changed between the campaigns as the contaminant concentrations varied. As there is no air injection in the instrument in this work, the type of pressure-dependent zero-decay rate such as observed by Mao et al. (2009) and Thames et al. (2020) does not apply.

## 3.5  Validation of the OH reactivity measurements using a propane gas standard

The accuracy of the measurements was tested by providing well-defined concentrations of propane in humidified synthetic air. The gas-mixture was prepared in a canister by mixing propane (purity $99.5\,\%$, Linde) with nitrogen. The resulting mixing ratio of $(2166\pm22)\,\mathrm{ppmv}$ was measured using the total organic carbon method described in detail in Berg et al. (2024), in which all carbon is converted to $CO_2$ on a heated palladium catalyst. The $CO_2$ is measured with a high accuracy using cavity ring-down

spectroscopy (Picarro, G1301) allowing to back-calculate the propane concentration.

For the tests with the OH reactivity instrument, a small flow of a few $\mathrm{ml/min}$ of this propane gas standard was mixed with a large flow of humidified synthetic air that was prepared in the same way as done for the zero measurements. The inlet of the instrument was overflowed with this well-defined mixture. Contaminations from other OH reactants are therefore not expected to affect the measurements. All flows were provided by calibrated mass flow controllers (Bronkhorst, El-Flow Series).

Figure 6 shows the measured OH reactivity, when the propane concentration was varied between $1\cdot10^{12}\,\mathrm{cm}^{-3}$ and $30\cdot10^{12}\,\mathrm{cm}^{-3}$ resulting in OH reactivity values between 1.5 and $34\,\mathrm{s}^{-1}$. Two set of measurements were performed at different pressures ($997\,\mathrm{hPa}$ and $600\,\mathrm{hPa}$) that were in the range of typical pressure values experienced during the flights of the AEROMMA campaign. Due to the dilution at lower pressure, propane concentrations and OH reactivity values were also lower in this set of measurements. The slope of a regression analysis gives the reaction rate constant of the OH reaction with

propane. The accuracy of the value includes the accuracy of the propane concentration in the flow tube of $1.4\,\%$, which takes into account the uncertainties in the flows and the propane concentration in the canister, and the statistical error of the slope of $2.7\,\%$. The total accuracy is therefore $4.1\,\%$.

The reaction rate coefficient of the OH reaction with propane has been investigated in several studies. Calculations using the Arrhenius expressions in the recommendations show excellent agreement within $8\,\%$ (IUPAC, Atkinson et al. (2006)) and $4\,\%$

(NASA-JPL, Burkholder et al. (2020)) with the values obtained from the OH reactivity measurements for both pressure values (Table 1). This demonstrates the high accuracy of the new instrument's OH reactivity measurements as shown for previous

 

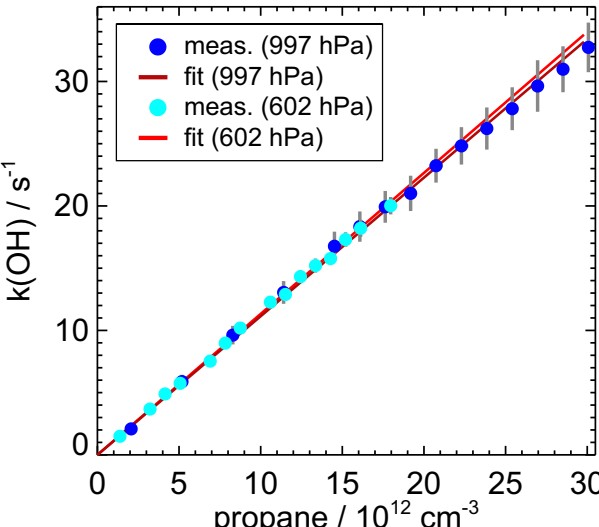

**Figure 6.** Determination of the bimolecular rate coefficient of the OH reaction with propane in air from OH reactivity measurements when the inlet of the instrument is overflowed with a mixture of propane in synthetic air. The rate coefficient can be described by an Arrhenius expression and is independent of pressure Atkinson et al. (2006); Burkholder et al. (2020). The rate coefficient is determined at two different values of pressure ($p$) to test the accuracy of measurements for the operational conditions on the aircraft.

**Table 1.** Reaction rate constant of the OH reaction with propane determined at two pressure values ($p$) by the OH reactivity instrument ($k_{meas}$) compared to recommendations by IUPAC ($k_{\mathrm{IUPAC}}$, Atkinson et al. (2006)) and NASA-JPL ($k_{\mathrm{NASA}}$, Burkholder et al. (2020)) at the respective temperature ($T$) of the measurement. The rate coefficient can be described by an Arrhenius expression and is independent of pressure. The rate coefficient is determined at two different values of pressure ($p$) to test the accuracy of measurements for the operational conditions on the aircraft.

| $p$ / hPa | $T$ / K | $k_{meas}$ / $10^{-12}\mathrm{cm^3s^{-1}}$ | $k_{\mathrm{IUPAC}}$ / $10^{-12}\mathrm{cm^3s^{-1}}$ | $k_{\mathrm{NASA}}$ / $10^{-12}\mathrm{cm^3s^{-1}}$ |
|---|---|---|---|---|
| 997 | 299 | $1.11 \pm 0.05$ | $1.08 \pm 0.2$ | $1.12 \pm 0.04$ |
| 602 | 296 | $1.13 \pm 0.05$ | $1.05 \pm 0.2$ | $1.09 \pm 0.04$ |





versions of the instrument (Lou et al., 2010; Berg et al., 2024). It is also shows that this high accuracy is also achieved for measurements on board of an aircraft when the sampled ambient air is at low pressure.

## 4  OH reactivity measurements on board the DC-8 aircraft during the AEROMMA campaign

The improved OH reactivity instrument was deployed on the NASA DC-8 aircraft during the AEROMMA campaign in summer 2023. The aircraft carried about 30 different instruments to measure gas-phase species and aerosol properties. Flights were conducted over the Pacific Ocean and over major urban areas in North America, including Chicago, Toronto, New York City, and Los Angeles. The detailed analysis of the measurements will be presented in separate papers. Here, two flights over the Pacific Ocean are shown to demonstrate the performance of the instrument when low reactivity values are observed and only

a few OH reactants are expected to contribute significantly to the OH reactivity, so that these measurements can demonstrate the high precision and accuracy of the measurements.

Figure 7 shows a map and the time series of the measured OH reactivity after subtracting the zero decay value. The measured OH reactivity and the value in ambient air outside the aircraft differ. One reason is the slightly different ($< 5\,\%$) number densities of the reactants in the flow tube, where the pressure is close to ambient pressure and the temperature is at cabin

temperature (295 to 305 K). The other reason is the possible influence of temperature and pressure on the OH rate constants.

The zero-decay rate is calculated from the parametrisation of Eq. 4, using the ambient humidity measurements from a diode laser hygrometer on board the aircraft and the pressure measurements in the flow tube of the OH reactivity instrument, resulting in values between 1.65 and $1.85\,\mathrm{s^{-1}}$. The OH reactivity from the OH reactants is calculated from various measurements listed in Table 2.

As would be expected in the clean, marine air, the OH reactivity is low with values between 1 and $1.5\,\mathrm{s^{-1}}$ over the Pacific Ocean and over land when the aircraft flew at high altitude. The measured OH reactivity can largely be explained by the presence of carbon monoxide (CO) and methane ($CH_4$), giving a reactivity of approximately $1\,\mathrm{s^{-1}}$. Individual contributions from other measured OH reactants are less than $0.2\,\mathrm{s^{-1}}$, of which formaldehyde and dimethyl sulfide are the largest.

The measurements demonstrate that the OH reactivity can be measured with high precision and accuracy on an aircraft

using the new instrument. The difference between measured total OH reactivity and calculations using individual reactants is in most cases less than $0.4\,\mathrm{s^{-1}}$, which is less than the accuracy of the measurement due to the uncertainty of the zero-decay rate. Differences are higher with values of up to $0.6\,\mathrm{s^{-1}}$ after the start of the flight and show a decreasing trend. This could be due to a slight drift in the value of the zero-decay rate, which could be caused by contamination if dirty air was sampled immediately after take-off leading to an increased OH wall loss.

At high altitudes (approximately $> 4\,\mathrm{km}$), the water vapour mixing ratio drops below 0.5 %, so that the initial OH concentration produced per photolysis laser shot is only around $1 \cdot 10^8\,\mathrm{cm^{-3}}$. At the same time, the expected OH reactivity becomes very low. Although up to 40 traces are summed before applying the fit procedure, resulting in an amplitude similar to traces acquired at higher water vapour mixing ratios, the scatter of the data is significantly increased (e.g. 17:30 to 18:00 21 June 2023, Fig. 7). At altitudes of more than $8\,\mathrm{km}$, the produced OH concentrations are too low to evaluate the OH decay.





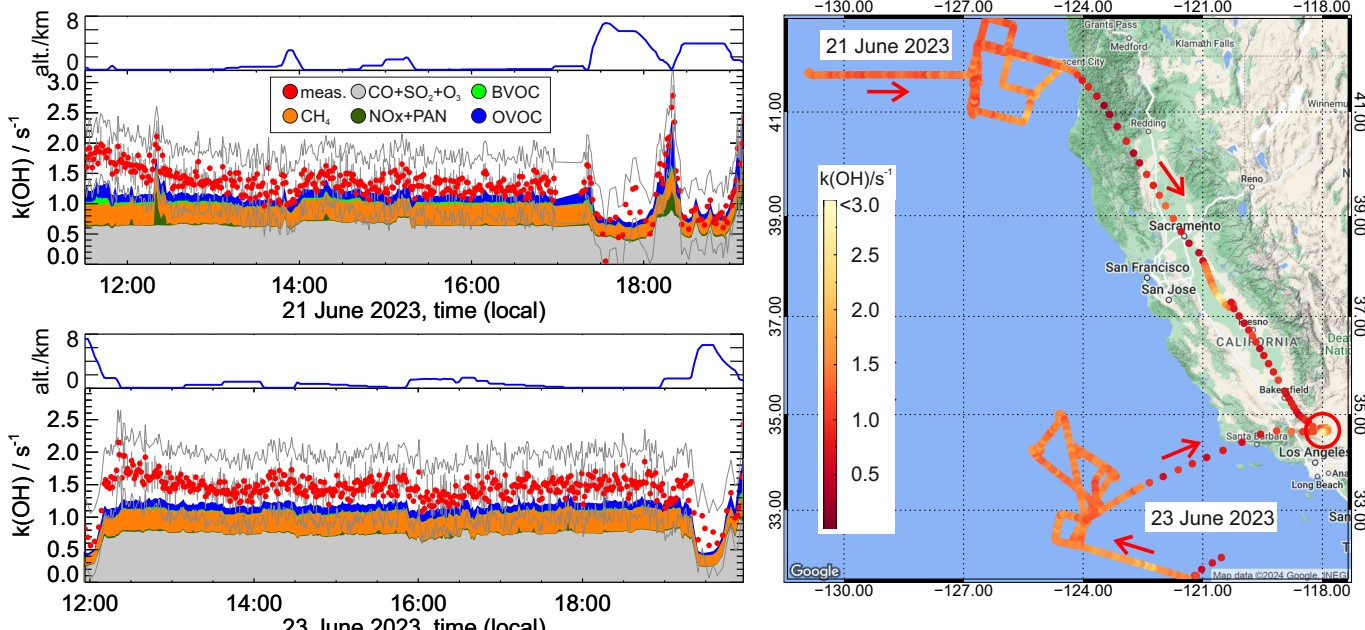

**Figure 7.** Time series and map of OH reactivity (1 min average) measured during the flights on 21 and 23 June 2023 over the Pacific Ocean starting from Palmdale, CA, USA (red circle). Coloured areas indicate contributions from measured OH reactants. Biogenic organic compounds (BVOCs) include dimethyl sulfide (DMS), isoprene, and monoterpenes and oxygenated organic compounds (OVOCs) include formaldehyde, acetaldehyde, ethanol, methyl vinyl ketone, methacrolein, nonanal, octanal, dimethyl sulfoxide (DMSO) and hydroperoxymethyl thioformate (HPMTF). The OH reactivity is given for conditions inside the flow tube of the instrument, where the pressure is close to ambient pressure, but the temperature is at cabin temperature (295 to 305 K). The zero decay value is subtracted. Error bars are the $1\sigma$ precision of measurements and the grey lines indicate the total uncertainty of the OH reactivity values of $0.5\,\mathrm{s}^{-1}$. The OH reactivity measurements started after the transit at high altitude (8 km), where water vapour mixing ratios were too low to produce sufficiently high OH concentrations in the flow tube.

Mao et al. (2009) provided the first OH reactivity measurements from an aircraft over marine environments during the INTEX-B measurement campaign in April 2006. An instrument was used, in which the OH decay was also measured directly by laser-induced fluorescence and in which a movable injector for radicals was used to vary the reaction time. The same group measured OH reactivity again with this instrument during the 4 ATom campaigns between 2016 and 2018 (Thames et al., 2020). The largest uncertainty in their measurements was due to the uncertainty in the zero decay value, as in the measurements in this work. In order to reduce the uncertainty, Mao et al. (2009) and Thames et al. (2020) adjusted the zero-decay rate so that

the measured reactivity agreed with calculations of OH reactants for certain parts of the flight at high altitudes over the oceans, assuming that there were no relevant unmeasured reactants in this clean air.

The OH reactivity observed during the INTEX-B and ATom campaigns over the oceans was maximum $2\,\mathrm{s}^{-1}$ and dropped to low values around $0.2\,\mathrm{s}^{-1}$ at high altitudes in the free troposphere. As during the AEROMMA campaign over the Pacific




**Table 2.** Instruments used for measurements shown in Fig. 7 or used to analyse OH reactivity data from the AEROMMA campaign.

| method | species | data version | instrument / reference |
| --- | --- | --- | --- |
| laser flash photolysis/laser-induced fluorescence | OH reactivity | R0[a] | this work |
| diode laser hygrometer | $H_2O$ | RA[a] | Diskin et al. (2002) |
| off-axis integrated cavity output spectroscopy | CO | R0[a] | LGR F-N2O/CO-23r, Bourgeois et al. (2022) |
| cavity ring-down spectroscopy | $CH_4$ | | Picarro 2401-m, Peischl et al. (2012) |
| chemiluminescence | $O_3$ | R0[a] | Ryerson et al. (1998) |
| laser-induced fluorescence | $SO_2$ | R0[a] | Rollins et al. (2016) |
| laser-induced fluorescence | $NO_2$, NO | R0[a] | Rollins et al. (2020) |
| laser-induced fluorescence | HCHO | R0[a] | Cazorla et al. (2015) |
| chemical ionisation mass spectrometry | peroxyacetyl nitrate (PAN), hydroperoxymethyl thioformate (HPMTF) | R0[a] R1[a] | Veres et al. (2020), Robinson et al. (2022) |
| chemical ionisation mass spectrometry | dimethyl sulfoxide (DMSO) | R0[a] | Xu et al. (2022) |
| proton-transfer-reaction mass spectrometry | dimethyl sulfide (DMS), monoterpenes, isoprene, methyl vinyl ketone + methacrolein, ethanol, nonanal, octanal acetaldehyde | R1[a] | Coggon et al. (2024) |

[a] https://csl.noaa.gov/projects/aeromma/data.html

Ocean, the OH reactivity was mainly due to CO and methane with small contributions from oxygenated organic compounds. Differences between the measured OH reactivity and calculations using OH reactant measurements were also similar and of the order of the uncertainty of the zero-decay rate. Thames et al. (2020) attempted to estimate the contribution of unmeasured species to the OH reactivity by a statistical approach and found that an OH reactivity between 0.4 and $0.7\,s^{-1}$ cannot be explained by measured OH reactants in the marine boundary layer.

## 5 Conclusions

An instrument for measuring OH reactivity using laser-flash photolysis and the direct detection of the OH decay by laser-induced fluorescence has been further developed in this work for use in field experiments in challenging environments such as on board of an aircraft. This instrument can operate largely autonomously and with a high sensitivity, providing a high precision of less than $0.3\,s^{-1}$ with a time resolution on the order of seconds. The accuracy of the measurements is mainly limited by the uncertainty in the zero-decay rate of $0.5\,s^{-1}$. Validation with a well-defined mixture of propane in synthetic air at two different pressures demonstrates that the measured OH reactivity values give OH reaction rate coefficients in excellent agreement with



values calculated from the Arrhenius expression recommended by IUPAC within 8 % (Atkinson et al., 2006) and NASA-JPL within 4 % (Burkholder et al., 2020).

The effort required to operate the instrument has been greatly reduced compared to previous versions. By simplifying the dye laser system used to detect the OH radicals, the instrument is robust against vibrations, and changes in the temperature and the pressure. If necessary, motorised mounts for the optical elements can automatically compensate for small changes in the laser alignment.

The good performance is demonstrated during the AEROMMA campaign, where measurements were conducted on board the NASA DC-8 aircraft. Measurements in a clean environments above the Pacific Ocean gave low OH reactivity values in the range 1.5 to $2.0\,\mathrm{s}^{-1}$. These low values are well explained by measured OH reactants with major contributions from carbon monoxide and methane within the uncertainty of the zero value that needs to be subtracted from the measurements.

Overall, the operational complexity of the new OH reactivity instrument could be significantly reduced compared to previous versions of the instrument and therefore has the potential for a wide application in laboratory and field experiments. Widespread use of OH reactivity measurements would provide valuable information on the load of pollutants in the atmosphere and the potential for the formation of secondary pollutants from their chemical transformation (Lelieveld et al., 2016).

*Data availability.* Data from the AEROMMA campaign from the NOAA data repository (https://csl.noaa.gov/projects/aeromma/data.html).

*Author contributions.* HF wrote the manuscript. AS, FB, AN, MF performed OH reactivity measurements and FB and RB performed TOC measurements. FH and AW contributed to the application of the instrument in the AEROMMA campaign. KB, SB, MC, GD, GG, CJ, JP, MR, AW, NS, RS, CS, PV, CW, EW, LX, KZ contributed to measurements used for the analysis of the measurements during the AEROMMA campaign. All co-authors discussed the content of the paper and contributed to the writing.

*Competing interests.* At least one of the (co-)authors is a member of the editorial board of Atmospheric Measurement Technique. The authors declare to have no other competing interests.

*Acknowledgements.* This research was in part supported by the Klaus Tschira Boost Fund, a joint initiative of the German Scholars Organization and the Klaus Tschira Stiftung and in part by NOAA cooperative agreement NA22OAR43200151. The authors thank the Department of Engineering and workshops at Forschungszentrum Jülich for the great support to develop the instrument. The authors also thank the entire team of the AEROMMA campaign for the support and opportunity to perform measurements during the campaign, specifically for the provision of additional data for comparing measured OH reactivity with calculations using single reactant concentrations (Formaldehyde: G. M. Wolfe, NASA Goddard Space Flight Center; PAN: G. Novak, NOAA).



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
