# Peer review of "Advances in an OH reactivity instrument for airborne field measurements"

_EGUsphere, 2024_

## Author Response (AR1)

**Responses to the comments of referee #1**

We thank the reviewer for the positive review of our paper.

**Responses to the comments of referee #2**

We thank the reviewer for the positive comments.

**Comment:** The paper compares the measured reactivity with that (I presume) calculated from the OH co-reactants which were measured (could be clearer). As the measurements cannot include all OH co-reactants the lifetime-derived reactivity should always be shorter (greater loss) – i.e. it should, by definition, disagree - rather than necessarily being an instrumental cause? Suggest the discussion is placed in this context.

**Response:** To better explain the calculations, we changed the text in Line 348: "The measured total OH reactivity is compared to calculations of the OH reactivity (Fig. 7) using OH reactant measurements from several instruments listed in Table 2. Measurements included inorganic compounds such as carbon monoxide (CO), sulfur dioxide ($SO_2$), nitrogen dioxide ($NO_2$), nitric oxide (NO), ozone ($O_3$) and organic compounds such as methane ($CH_4$) and formaldehyde (HCHO)." We agree that it is likely that not all OH reactants were measured. Nevertheless, the measurements show that the contribution of unmeasured reactants is less than the accuracy of the total OH reactivity measurement. In our opinion, the slightly larger discrepancies observed after take-off rather points to a small drift of the instrumental zero that could be caused by small contaminations in the flow tube when very polluted air is sampled. In contrast, it appears unlikely that the contribution of unmeasured OH reactants is higher during this period than during the rest of the flight when chemical conditions were presumably very similar. We added in Line 355: "Due to the large number of OH reactants in the atmosphere, the total OH reactivity measurements are expected to be rather higher than the calculations using the limited number of OH reactant measurements."

**Comment:** Similarly, there is only very passing description of the comparison of the measured levels with those inferred from CO/CH4 (other species ?) in the AEROMMA campaign.

**Response:** The contributions of CO and CH4 to the OH reactivity were highest among the OH reactants measured (Line 351). We added the mixing ratios of the species mentioned in the same paragraph. All species that are included in the calculations are listed in Table 2.

**Comment:** The paper would benefit from some discussion of this approach to understanding reactivity measurements vs the species which can possibly be measured. Reference to the full degradation (as predicted by e.g. GECKO-A) might be worth including (not the chemical detail – the principle). I appreciate that the point of this paper is not to perform a full chemical analysis of the campaign data, but this would help place the instrument in wider science context.

**Response:** A full analysis of the chemical degradation of the reactants is indeed beyond the scope of this paper. However, the largest contributions to the OH reactivity are from CO and methane, which major oxidation products are either included in the measurements (formaldehyde) or which do not react with OH (CO2). Contributions of more complex, measured

hydrocarbons are very small so that also only small contributions are expected from their oxidation products. Therefore, we do not expect that a chemical box model calculation would reduce the small gap between measured and calculated OH reactivity. We added in Line 357: "Since the reactivity of non-methane hydrocarbons was very small, the contribution of unmeasured oxygenated hydrocarbons that could be expected from their oxidation was likely to be small."

**Comment:** Title – currently suggests the paper will in some way review or compare different instruments for OH reactivity, while in reality only one system is considered– suggest to fine tune the wording.

**Response:** We change the title to "Advances in an OH reactivity instrument for airborne field measurements".

**Comment:** Abstract L6 – no secondary chem impacts for typical atmospheric conditions – is this true for the high NO of e.g. some polluted megacities?

**Response:** There are conditions, where high NO can impact the measurements, but these are exceptionally high concentrations, which are typically only observed very close to sources like in tunnels or street canyons. Tests and a detailed discussion can be found in our comparison of OH reactivity instruments (Fuchs et al., AMT, 2017) and in the characterisation of an earlier version of the instrument (Lou et al., ACP, 2010). A summary is included in this manuscript in L143 to L155.

**Comment:** L23 not *all* inorganic components are chemically transformed

**Response:** We cancelled "all".

**Comment:** L24 make clear talking about primary OH production (rather than e.g. HO2 + NO)

**Response:** We modified the sentence to "... the hydroxyl radical (OH), which is formed primarily by ..."

**Comment:** L31 secondary pollutants – through this route (as you're not sensitive to secondary species formation through O3, NO3 etc)

**Response:** We agree with this statement, although most of the reactants that react with OH also react with O3 and NO3 and the rate coefficients often scale with the OH rate coefficients. Therefore, OH reactivity can often be used as a proxy for the potential for secondary pollutant formation when the oxidation is happening by other oxidants. We added "from OH oxidation".

**Comment:** L50/51 – might be clearer to phrase as OH "co-reactant" consumed

**Response:** We changed the text accordingly.

**Comment:** L110 illumination -how much is "almost" the full cross section – quantify?

**Response:** We specified the diameter to be 30mm in L108.

**Comment:** L280ish – it would be useful to discuss possible impacts of HO2 + NO recycling on kOH in heavily polluted atmospheres – approach to correction if needed / implications for deployment across all atmospheric regimes.

**Response:** We do not see a potential effect of HO2+NO recycling on the OH reactivity measurement as part of the zero value characterisation and would not include this in the section suggested by the reviewer. We feel that our discussion of this effect in  L143 to L155 addresses the points raised by the reviewer.

**Comment:** L317 is calibration checked in the field or is this just a lab procedure

**Response:** This is only a laboratory characterisation measurement. We believe that this is sufficient to determine the accuracy of the measurements, as the fundamental properties of the instrument should not change when the instrument is used in the field. In our experience, most changes in the instrument performance are visible as a deviation of the observed OH decay from a single exponential behaviour. This is checked for all decays measured in the field.

**Comment:** Fig 6 – just a hint of a curve in the propane data at the higher concentrations? Do reactions of OH with C3H6 degradation products become significant?

**Response:** We can exclude that degradation products affect the observed OH loss as the initial OH concentration was much lower than the propane concentration so that there is no significant contribution to the OH reactivity. The slight curvature of the is very small in absolute terms. The statistical error is greater than the apparent curvature so that we do not consider this to be significant.

**Comment:** L397 complexity has been reduced

**Response:** We changed the text accordingly.